# Diversify to Generalize: Learning Generalized Representations for Time Series Classification

## Abstract

Time series classification is an important problem in real world. Due to its non-stationary property that the distribution changes over time, it remains challenging to build models for generalization to unseen distributions. In this paper, we propose to view the time series classification problem from the *distribution* perspective. We argue that the temporal complexity attributes to the unknown latent distributions within. To this end, we propose DIVERSIFY to learn generalized representations for time series classification. DIVERSIFY takes an iterative process: it first obtains the worst-case distribution scenario via adversarial training, then matches the distributions between all segments. We also present some theoretical insights. Extensive experiments on gesture recognition, speech commands recognition, and sensor-based human activity recognition demonstrate that DIVERSIFY significantly outperforms other baselines and effectively characterizes the latent distributions by qualitative and quantitative analysis.

## 1 Introduction

Time series classification is one of the most challenging problems in machine learning and statistics community. Example applications include sensor-based human activity recognition, Parkinson's disease diagnosis, and electronic power consumption (Fawaz et al., 2019). One important nature of time series is non-stationary property, which means that its statistical features are changing over time. For years, there have been tremendous efforts to tackle the time series classification problem, such as hidden Markov models (Fulcher & Jones, 2014), RNN-based methods (Hüsken & Stagge, 2003), and Transformer-based approaches (Li et al., 2019).

In this paper, we are specifically interested in modeling time series from the *distribution* perspective. More precisely, we aim to learn representations for time series that can generalize to *unseen* distributions. Note that this scenario has been extensively studied in existing literature of domain generalization (Muandet et al., 2013; Wang et al., 2021a) and out-of-distribution generalization (Krueger et al., 2021), where researchers are keen to bridge the gap between known and unknown distributions, thus generalize well. While most of the efforts are done in image classification, few of them focus on the time series domain, which is more challenging. Although time series share a similar goal as image data in domain generalization, it naturally brings more challenges due to its non-stationary property: the distribution keeps changing over time, which contains diverse distribution information that should be harnessed well for better generalization.

We show an illustrative example in Figure 1. Domain generalization in image classification often involves several domains and the domain information is known (subfigure (a)). Thus, we can leverage such domain information to build generalization models. However, in Figure 1 (b), we see that in time series data, although the distribution is changing dynamically over time, its domain information is not available. This will dramatically impede the modeling of existing domain generalization algorithms as they typically assume access to domain information (subfigure (c)).

In order to learn a generalized time series model, we propose DIVERSIFY, a domain generalization algorithm to characterize the latent distributions inside the time series data. Concretely speaking, our method consists of a min-max adversarial game that: on one hand, it learns to segment the time series data into several latent sub-domains by maximizing the segment-wise distribution gap to preserve

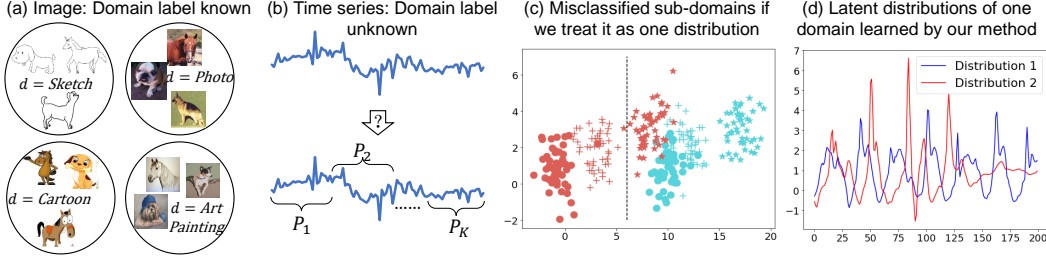

Figure 1: Illustration of DIVERSIFY: (a) Domain generalization for image data require known domain labels. (b) Domain label is unknown for time series. (c) If we treat the time series data as one single domain, the sub-domains are misclassified. Different colors and shapes correspond to different classes and domains. (d) Finally, our DIVERSIFY can effectively learn the latent distributions.

diversities, i.e., the *worst-case* distribution scenario; on the other hand, it learns domain-invariant representations by reducing the distribution divergence for the worst-case scenario. Such diversification naturally exists in a non-stationary dataset where the data from multiple people naturally follow several latent distributions. Moreover, it is also surprising to find that even the data of one person still has such diversification: it can also be split into several latent distributions. Obviously, DIVERSIFY can effectively characterize the latent distributions (Figure 1 (d)).

To summarize, our contributions are three-fold:

- Novel problem: For deep learning-based time series classification, we identify the generalized representation problem, which is challenging than traditional image classification problem due to the existence of unidentified latent distributions.
- New methodology: We propose DIVERSIFY, a theoretically-motivated solution to solve the generalized representation learning problem to identify the latent distributions.
- Good performance: Our approach is extensively evaluated in three types of tasks: gesture recognition, speech command recognition, and sensor-based activity recognition. By qualitative and quantitative analysis, we demonstrate the superiority of DIVERSIFY on several challenging scenarios: on difficult tasks, significantly diverse datasets, and limited data.

## 2 METHODOLOGY

### 2.1 PROBLEM FORMULATION

We are given a time-series dataset $\mathcal{D}^{tr} = \{(\mathbf{x}_i, y_i)\}_{i=1}^N$ as the training dataset, where $N$ is the number of samples, $\mathbf{x}_i \in \mathcal{X} \subset \mathbb{R}^p$ is the $p$-dimensional instance (sliding window) and $y_i \in \mathcal{Y} = \{1, \ldots, C\}$ is its label. We use $\mathbb{P}^{tr}(\mathbf{x}, y)$ on $\mathcal{X} \times \mathcal{Y}$ to denote the joint distribution of the training dataset. Our goal is to learn a generalized model from $\mathcal{D}^{tr}$ to predict well on an *unseen* target dataset, $\mathcal{D}^{te}$, which is inaccessible in training. Like $\mathcal{D}^{tr}$, time series in $\mathcal{D}^{te}$ also be split into short series. In our problem, the training and test datasets have the same input and output spaces but different distributions, i.e., $\mathcal{X}^{tr} = \mathcal{X}^{te}$, $\mathcal{Y}^{te} = \mathcal{Y}^{te}$, but $\mathbb{P}^{tr}(\mathbf{x}, y) \neq \mathbb{P}^{te}(\mathbf{x}, y)$. We aim to train a model $h$ from $\mathcal{D}^{tr}$ to minimize the risk on $\mathcal{D}^{te}$: $\min_h \mathbb{E}_{(\mathbf{x}, y) \sim \mathbb{P}^{te}}[h(\mathbf{x}) \neq y]$.

Note that due to the non-stationary property, the training dataset may be composed of several unknown latent distributions, instead of one fixed distribution, i.e., $\mathbb{P}^{tr}(\mathbf{x}, y) = \sum_{i=1}^K \pi_i \mathbb{P}^i(\mathbf{x}, y)$, where $\mathbb{P}^i(\mathbf{x}, y)$ is the distribution of the $i$-th sub-domains of the training data and $\pi_i$ is its weight. $K$ is the number of sub-domains that is unknown, and $\sum_{i=1}^K \pi_i = 1$.

### 2.2 MOTIVATION

The labeled time series data can be composed of several latent distributions (domains) that are challenging to characterize, even if the dataset is fully labeled. For instance, data collected by sensors of

three persons may belong to two different distributions when considering their similarities. Moreover, even for data from one single person, different segments of one sequence may follow different distributions. In a nutshell, in reality, there often exist several sub-domains in one time series dataset.

To ensure good generalization performance on the test dataset, it is important to learn *distribution-invariant*, or domain-invariant representations from the training dataset by characterizing its *latent* distributions. These latent distributions may contain both benign and malignant knowledge that influences generalization on the target dataset. In Figure 1 (c), we assume the source domain contains two sub-domains (circle and plus points). Directly learning from the entire source domain by treating it as one distribution may generate the black margin. Although green star data points and red star points can be classified easily, red star data points are misclassified to the green class when predicting on the out-of-distribution domain (star points) with the learned model. While other methods fail, our method can characterize the latent distributions, which will be introduced later.

## 2.3 DIVERSIFY

In this paper, we propose DIVERSIFY to learn generalized representations for time series classification. The core of DIVERSIFY is to characterize the latent distributions in a time series dataset and then to minimize the distribution divergence between each two. To characterize the diverse information for better generalization, DIVERSIFY utilizes an iterative process: it first obtains the worst-case distribution scenario from a given dataset, then bridges the distribution gaps between each pair of latent distributions. Why the worst-case scenario? We argue that the worst-case scenario will maximally preserve the diverse information of each latent distribution, thus benefiting generalization.

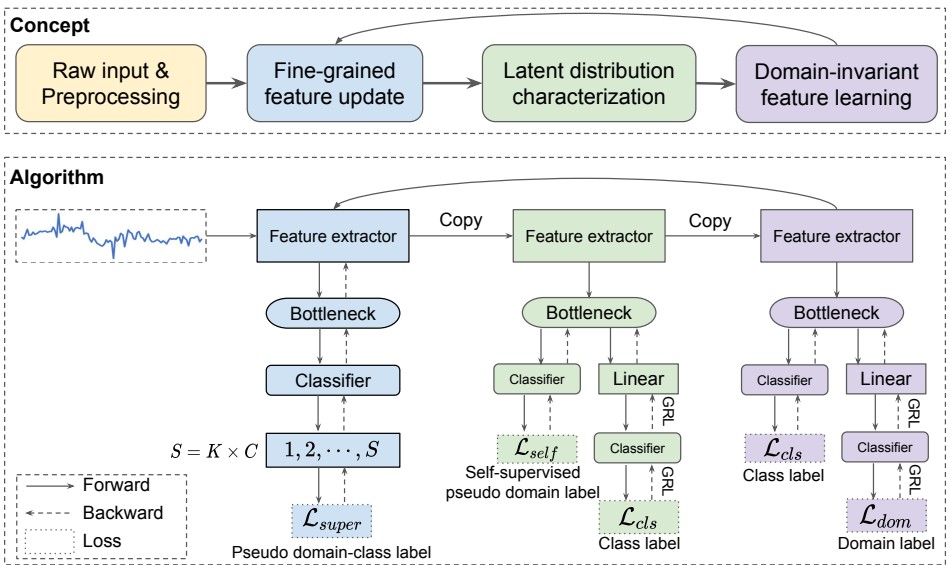

Figure 2: The framework of DIVERSIFY.

Figure 2 describes the main procedures of DIVERSIFY:

1. Pre-processing: this step adopts the sliding window to split the entire training dataset into fixed-size windows. We believe that the data from one window is the smallest domain unit.

2. Fine-grained feature update: this step updates the feature extractor using the proposed *pseudo domain-class* labels as the supervision.

3. Latent distribution characterization: this step aims to identify the domain label for each instance to acquire the latent distribution information. It tries to *maximize* the different distribution gaps to enlarge diversity.

4. Domain-invariant representation learning: this step utilizes pseudo domain labels from the last step to learn domain-invariant representations and train a generalizable model.

Note that steps $2 \sim 4$ are iterative as shown in Figure 2. Now we elaborate their details.

**Fine-grained Feature Update**    Before we characterize the latent distributions, we perform fine-grained feature update to obtain useful representations. As shown in Figure 2 (blue), in order to fully utilize the knowledge contained in the domains and classes, we propose a new concept: *pseudo domain-class label*, which serves as the supervision for feature extractor. This will ensure that the feature extractor can learn fine-grained information which benefits the later steps.

At the first iteration, there is no domain label $d'$ and we simply initialize $d' = 0$ for all samples. We treat per class per domain as a new class, and set the label to $s \in \{1, 2, \cdots, S\}$. We have $S = K \times C$ where $K$ is the pre-defined number of latent distributions that can be tuned in experiments. We perform the following pseudo domain-class label assignment to get discrete values for supervision: $s = d' \times C + y$. (Response to reviewer kiW8: this equation is put inline.)

Let $h_f^{(2)}, h_b^{(2)}, h_c^{(2)}$ be feature extractor, bottleneck, and classifier, respectively (we use superscript to denote step number), then, the supervised loss is computed using the cross-entropy loss $\ell$:

$$\mathcal{L}_{super} = \mathbb{E}_{(\mathbf{x},y) \sim \mathbb{P}^{tr}} \ell \left( h_c^{(2)}(h_b^{(2)}(h_f^{(2)}(\mathbf{x}))), s \right). \tag{1}$$

**Latent Distribution Characterization**    This step characterizes the latent distributions contained in one dataset. As shown in Figure 2 (green), we employ an adversarial training strategy to disentangle the domain labels from the class labels. However, there are no actual domain labels provided, which hinders such disentanglement. Inspired by (Liang et al., 2020), we adopt a *self-supervised pseudo-labeling* strategy to obtain domain labels.

Firstly, we attain the centroid for each domain with class-invariant features:

$$\tilde{\mu}_k = \frac{\sum_{\mathbf{x}_i \in \mathcal{X}^{tr}} \delta_k(h_c^{(3)}(h_b^{(3)}(h_f^{(3)}(\mathbf{x}_i)))) h_b^{(3)}(h_f^{(3)}(\mathbf{x}_i))}{\sum_{\mathbf{x}_i \in \mathcal{X}^{tr}} \delta_k(h_c^{(3)}(h_b^{(3)}(h_f^{(3)}(\mathbf{x}_i))))}, \tag{2}$$

where $h_f^{(3)}, h_b^{(3)}, h_c^{(3)}$ are feature extractor, bottleneck, and classifier, respectively. $\tilde{\mu}_k$ is the initial centroid of the $k^{th}$ latent sub-domains while $\delta_k$ is the $k^{th}$ element of the logit soft-max output. Then, we obtain the pseudo domain labels via the nearest centroid classifier using a distance function $D$:

$$\tilde{d}'_i = \arg \min_k D(h_b^{(3)}(h_f^{(3)}(\mathbf{x}_i)), \tilde{\mu}_k). \tag{3}$$

Then, we compute the centroids based on the new pseudo labels and obtain the updated pseudo domain labels:

$$\mu_k = \frac{\sum_{\mathbf{x}_i \in \mathcal{X}^{tr}} \mathbb{I}(\tilde{d}'_i = k) h_b^{(3)}(h_f^{(3)}(\mathbf{x}))}{\sum_{\mathbf{x}_i \in \mathcal{X}^{tr}} \mathbb{I}(\tilde{d}'_i = k)},$$
$$d'_i = \arg \min_k D(h_b^{(3)}(h_f^{(3)}(\mathbf{x}_i)), \mu_k), \tag{4}$$

where $\mathbb{I}(a) = 1$ when $a$ is true, otherwise 0. After we obtain $d'$, we can compute the goal of step 2:

$$\mathcal{L}_{self} + \mathcal{L}_{cls} = \mathbb{E}_{(\mathbf{x},y) \sim \mathbb{P}^{tr}} \ell(h_c^{(3)}(h_b^{(3)}(h_f^{(3)}(\mathbf{x}))), d') + \ell(h_{adv}^{(3)}(R_{\lambda_1}(h_b^{(3)}(h_f^{(3)}(\mathbf{x})))), y), \tag{5}$$

where $h_{adv}^{(3)}$ is the discriminator for step 3 that contains several linear layers and one classification layer. $R_{\lambda_1}$ is the gradient reverse layer with hyperparameter $\lambda_1$ (Ganin et al., 2016). After this step ends, we can obtain pseudo domain label $d'$ for $\mathbf{x}$.

**Domain-invariant Representation Learning**    After obtaining the latent distributions, we learn domain-invariant representations for generalization. In fact, this step (Figure 2 purple) is simple: we borrow the idea from DANN (Ganin et al., 2016) and directly use adversarial training to update the classification loss $\mathcal{L}_{cls}$ and domain classifier loss $\mathcal{L}_{dom}$ using GRL:

$$\mathcal{L}_{cls} + \mathcal{L}_{dom} = \mathbb{E}_{(\mathbf{x},y) \sim \mathbb{P}^{tr}} \ell(h_c^{(4)}(h_b^{(4)}(h_f^{(4)}(\mathbf{x}))), y) + \ell(h_{adv}^{(4)}(R_{\lambda_2}(h_b^{(4)}(h_f^{(4)}(\mathbf{x})))), d'), \tag{6}$$

where $\ell$ is the cross-entropy loss and $R_{\lambda_2}$ is the gradient reverse layer with hyperparameter $\lambda_2$ (Ganin et al., 2016). More details of GRL and adv. training are presented in appendix A.1.

We repeat the above three steps until convergence or max epochs. The final model is selected via a validation dataset split from the source domain (Gulrajani & Lopez-Paz, 2021).

As for inference, we predict the labels with the modules from the last step.

## 2.4 THEORETICAL INSIGHTS

We present some theoretical insights to show that our approach is well motivated in theory.

**Proposition 2.1.** *(Proposition 3.1 in (Sicilia et al., 2021)) Let $\mathcal{X}$ be a space and let $\mathcal{H}$ be a class of hypotheses corresponding to this space. Let $\mathbb{Q}$ and the collection $\{\mathbb{P}_i\}_{i=1}^k$ be distributions over $\mathcal{X}$ and let $\{\varphi_i\}_{i=1}^k$ be a collection of non-negative coefficients with $\sum_i \varphi_i = 1$. Let the object $\mathcal{O}$ be a set of distributions such that for every $\mathbb{S} \in \mathcal{O}$ the following holds*

$$\sum_i \varphi_i d_{\mathcal{H}\Delta\mathcal{H}}(\mathbb{P}_i, \mathbb{S}) \leq \max_{i,j} d_{\mathcal{H}\Delta\mathcal{H}}(\mathbb{P}_i, \mathbb{P}_j). \tag{7}$$

*Then, for any $h \in \mathcal{H}$, the following equality holds:*

$$\varepsilon_{\mathbb{Q}}(h) \leq \lambda_\varphi + \sum_i \varphi_i \varepsilon_{\mathbb{P}_i}(h) + \frac{1}{2}\min_{\mathbb{S}\in\mathcal{O}} d_{\mathcal{H}\Delta\mathcal{H}}(\mathbb{S}, \mathbb{Q}) + \frac{1}{2}\max_{i,j} d_{\mathcal{H}\Delta\mathcal{H}}(\mathbb{P}_i, \mathbb{P}_j), \tag{8}$$

*where $\lambda_\varphi = \sum_i \varphi_i \lambda_i$ and each $\lambda_i$ is the error of an ideal joint hypothesis for $\mathbb{Q}$ and $\mathbb{P}_i$. $\varepsilon_{\mathbb{P}}(h)$ is the error for a hypothesis $h$ on a distribution $\mathbb{P}$. $d_{\mathcal{H}\Delta\mathcal{H}}(\mathbb{P}, \mathbb{Q})$ is $\mathcal{H}$-divergence which measure differences in distribution (Ben-David et al., 2010).*

The first item in Eq. 8, $\lambda_\varphi$, is often neglected since it is small in reality. The second item, $\sum_i \varphi_i \varepsilon_{\mathbb{P}_i}(h)$, exists in almost all methods and can be minimized via supervision from class labels with cross-entropy loss in Eq. 6. Our main purpose is to minimize the last two items in Eq. 8. Here $\mathbb{Q}$ corresponds to the unseen out-of-distribution target domain.

The last term $\frac{1}{2}\max_{i,j} d_{\mathcal{H}\Delta\mathcal{H}}(\mathbb{P}_i, \mathbb{P}_j)$ is common in DA and DG which measures the maximum differences among source domains. This corresponds to step 4 in our approach.

Finally, the third item, $\frac{1}{2}\min_{\mathbb{S}\in\mathcal{O}} d_{\mathcal{H}\Delta\mathcal{H}}(\mathbb{S}, \mathbb{Q})$, explains why we exploit sub-domains in step 3. Since our goal is to learn a model which can perform well on an unseen target domain, we cannot obtain $\mathbb{Q}$. To minimize $\frac{1}{2}\min_{\mathbb{S}\in\mathcal{O}} d_{\mathcal{H}\Delta\mathcal{H}}(\mathbb{S}, \mathbb{Q})$, we can only enlarge the range of $\mathcal{O}$. We have to $\max_{i,j} d_{\mathcal{H}\Delta\mathcal{H}}(\mathbb{P}_i, \mathbb{P}_j)$ according to Eq. 7, corresponding to step 3 in our method which tries to segment the time series data into several latent sub-domains by maximizing the segment-wise distribution gap to preserve diversities, i.e., the worst-case distribution scenario.

## 3 EXPERIMENTS

We extensively evaluate our approach in three time series classification tasks: gesture recognition, speech commands recognition, and sensor-based human activity recognition. These applications represent diverse situations of time series, thus can thoroughly reflect the advantage of our method. We will introduce their detail later. Per-segment accuracy is adopted as the evaluation metric.

For all experiments, we conduct the training-domain-validation strategy following DomainBed (Gulrajani & Lopez-Paz, 2021). The training data are split by $8:2$ for training and validation. For comparison, we re-implement eight recent strong comparison methods following DomainBed: ERM, DANN (Ganin et al., 2016), CORAL (Sun & Saenko, 2016), Mixup (Zhang et al., 2018), GroupDRO (Sagawa et al., 2020), RSC (Huang et al., 2020), ANDMask (Parascandolo et al., 2021), and GILE (Qian et al., 2021). For fairness, all methods (except GILE) utilize a feature net with two blocks and each block has one convolution layer, one pooling layer, and one batch normalization layer. Moreover, the bottleneck and classification layer adopts a FC layer, respectively. See appendix C for more details. Note that most methods require the domain labels known in training while our does not, which is more challenging and practical. We tune all methods to report the averaged best performance of three trials.

### 3.1 GESTURE RECOGNITION

Electromyography (EMG) is a common type of time-series data that is based on bioelectric signals. We use EMG for gestures Data Set (Lobov et al., 2018) that contains raw EMG data recorded by MYO Thalmic bracelet. The bracelet is equipped with eight sensors equally spaced around the forearm that simultaneously acquire myographic signals. Data of 36 subjects are collected while they

performed series of static hand gestures and the number of instances is $40,000 - 50,000$ recordings in each column. It contains 7 classes and we select 6 common classes for our experiments. We randomly divide 36 subjects into four domains without overlapping and each domain contains data of 9 persons. Experiments are done in three random trials, thus mitigating influence of unfair splits.

Since EMG data comes from bioelectric signals, it is affected by many factors. EMG data are scene and device-dependent, which means the same person may generate different data when performing the same activity with the same device at a different time or with the different devices at the same time. Therefore, the benchmark of EMG is challenging. Table 1 shows that our method achieves the best average performance and is $4.3\%$ better than the second-best method. Table 7 in appendix shows the advantage of our method in different model sizes.

Table 1: Results on EMG dataset.

| Target | 0 | 1 | 2 | 3 | AVG |
|---|---|---|---|---|---|
| ERM | 62.6 | 69.9 | 67.9 | 69.3 | 67.4 |
| DANN | 62.9 | 70.0 | 66.5 | 68.2 | 66.9 |
| CORAL | 66.4 | 74.6 | 71.4 | 74.2 | 71.7 |
| Mixup | 60.7 | 69.9 | 70.5 | 68.2 | 67.3 |
| GroupDRO | 67.6 | 77.4 | 73.7 | 72.5 | 72.8 |
| RSC | 70.1 | 74.6 | 72.4 | 71.9 | 72.2 |
| ANDMask | 66.5 | 69.1 | 71.4 | 68.9 | 69.0 |
| DIVERSIFY | **71.7** | **82.4** | **76.9** | **77.3** | **77.1** |

## 3.2 SPEECH COMMANDS

Then, we adopt a regular speech recognition task, the Speech Commands dataset (Warden, 2018). The dataset consists of one-second audio recordings of both background noise and spoken words such as 'left', 'right', etc. It is collected from more than 2,000 persons, thus is more complicated. Following (Kidger et al., 2020), we use 34,975 time series corresponding to ten spoken words to produce a balanced classification problem. Since this dataset is collected from multiple persons, thus the training and test distributions are different, which is also an OOD problem with one training domain. There are many subjects and each subject only records few audios. Thus, an audio recording is short so we do not split each sample.

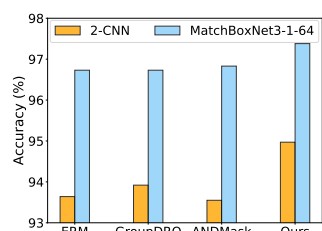

Figure 3: Results on Speech commands dataset with two different backbones.

Figure 3 shows the recognition results on two backbones. Compared with GroupDRO, DIVERSIFY has over $1\%$ improvement based on a basic CNN architecture and over $0.6\%$ improvement based on MatchBoxNet3-1-64 (Majumdar & Ginsburg, 2020) (its baseline is already strong). The results demonstrate the superiority of our method on a regular time-series benchmark composed of massive distributions.

## 3.3 SENSOR-BASED HUMAN ACTIVITY RECOGNITION

Finally, we employ several popular sensor-based human activity recognition datasets for more detailed evaluation. UCI daily and sports dataset (**DSADS**) (Barshan & Yüksek, 2014) consists of 19 activities collected from 8 subjects wearing body-worn sensors on 5 body parts. USC-SIPI human activity dataset (**USC-HAD**) (Zhang & Sawchuk, 2012) is composed of 14 subjects (7 male, 7 female, aged from 21 to 49) executing 12 activities with a sensor tied on the front right hip. **UCI-HAR** (Anguita et al., 2012) is collected by 30 subjects performing 6 daily living activities with a waist-mounted smartphone. **PAMAP** physical activity monitoring dataset (Reiss & Stricker, 2012) contains data of 18 activities, performed by 9 subjects wearing 3 sensors.

We construct *four* different settings representing different degrees of generalization: (1) **Cross-person generalization**: This setting utilizes DSADS, USC-HAD, PAMAP[1] datasets to construct three benchmarks. Within each dataset, we randomly split the data into four groups[2] and then use three groups as training data to learn a generalized model for the last group. (2) **Cross-position generalization**: this setting uses DSADS dataset and data from each position denotes a different domain. Each sample contains three sensors with nine dimensions. We treat one position as the test domain while the others are for training. (3) **Cross-dataset generalization**: this setting uses all four datasets, and each dataset corresponds to a different domain. Six common classes are selected. Two sensors from each dataset that belong to the same position are selected and data is down-sampled to ensure the same dimension of data. (4) **One-Person-To-Another**. This setting adopts DSADS,

---

[1]We do not use UCI-HAR in cross-person setting since its baseline is good enough.

[2]For more details on dataset information and domain splits, please refer to Appendix B.

Table 2: Classification accuracy on cross-person generalization.

| Target | DSADS | | | | | USC-HAD | | | | | PAMAP | | | | | ALL |
|---|---|---|---|---|---|---|---|---|---|---|---|---|---|---|---|---|
| | 0 | 1 | 2 | 3 | AVG | 0 | 1 | 2 | 3 | AVG | 0 | 1 | 2 | 3 | AVG | AVG |
| ERM | 83.1 | 79.3 | 87.8 | 71.0 | 80.3 | 81.0 | 57.7 | 74.0 | 65.9 | 69.7 | 90.0 | 78.1 | 55.8 | 84.4 | 77.1 | 75.7 |
| DANN | 89.1 | 84.2 | 85.9 | 83.4 | 85.6 | 81.2 | 57.9 | 76.7 | 70.7 | 71.6 | 82.2 | 78.1 | 55.4 | 87.3 | 75.7 | 77.7 |
| CORAL | 91.0 | 85.8 | 86.6 | 78.2 | 85.4 | 78.8 | 58.9 | 75.0 | 53.7 | 66.6 | 86.2 | 77.8 | 49.0 | 87.8 | 75.2 | 75.7 |
| Mixup | 89.6 | 82.2 | 89.2 | 86.9 | 87.0 | 80.0 | 64.1 | 74.3 | 61.3 | 69.9 | 89.4 | 80.3 | 58.4 | 87.7 | 79.0 | 78.6 |
| GroupDRO | 91.7 | 85.9 | 87.59 | 78.3 | 85.9 | 80.1 | 55.5 | 74.7 | 60.0 | 67.6 | 85.2 | 77.7 | 56.2 | 85.0 | 76.0 | 76.5 |
| RSC | 84.9 | 82.3 | 86.7 | 77.7 | 82.9 | 81.9 | 57.9 | 73.4 | 65.1 | 69.6 | 87.1 | 76.9 | 60.3 | 87.8 | 78.0 | 76.9 |
| ANDMask | 85.0 | 75.8 | 87.0 | 77.6 | 81.4 | 79.9 | 55.3 | 74.5 | 65.0 | 68.7 | 86.7 | 76.4 | 43.6 | 85.6 | 73.1 | 74.4 |
| GILE | 81.0 | 75.0 | 77.0 | 66.0 | 74.7 | 78,0 | 62.0 | 77.0 | 63.0 | 70.0 | 83.0 | 68.0 | 42.0 | 76.0 | 67.5 | 70.7 |
| DIVERSIFY | 90.4 | 86.5 | 90.0 | 86.1 | 88.2 | 82.6 | 63.5 | 78.7 | 71.3 | 74.0 | 91.0 | 84.3 | 60.5 | 87.7 | 80.8 | 81.0 |

Table 3: Classification accuracy on cross-position, cross-dataset, and one-to-another generalization.

| Target | Cross-position generalization | | | | | | Cross-dataset generalization | | | | | One-Person-To-Another | | | |
|---|---|---|---|---|---|---|---|---|---|---|---|---|---|---|---|
| | 0 | 1 | 2 | 3 | 4 | AVG | 0 | 1 | 2 | 3 | AVG | DSADS | USC-HAD | PAMAP | AVG |
| ERM | 41.5 | 26.7 | 35.8 | 21.4 | 27.3 | 30.6 | 26.4 | 29.6 | 44.4 | 32.9 | 33.3 | 51.3 | 46.2 | 53.1 | 50.2 |
| DANN | 45.4 | 25.3 | 38.1 | 28.9 | 25.1 | 32.6 | 29.7 | 45.3 | 46.1 | 43.8 | 41.2 | - | - | - | - |
| CORAL | 33.2 | 25.2 | 25.8 | 22.3 | 20.6 | 25.4 | 39.5 | 41.8 | 39.1 | 36.6 | 39.2 | - | - | - | - |
| Mixup | 48.8 | 34.2 | 37.5 | 29.5 | 29.9 | 36.0 | 37.3 | 47.4 | 40.2 | 23.1 | 37.0 | 62.7 | 46.3 | 58.6 | 55.8 |
| GroupDRO | 27.1 | 26.7 | 24.3 | 18.4 | 24.8 | 24.3 | 51.4 | 36.7 | 33.2 | 33.8 | 38.8 | 51.3 | 48.0 | 53.1 | 50.8 |
| RSC | 46.6 | 27.4 | 35.9 | 27.0 | 29.8 | 33.3 | 33.1 | 39.7 | 45.3 | 45.9 | 41.0 | 59.1 | 49.0 | 59.7 | 55.9 |
| ANDMask | 47.5 | 31.1 | 39.2 | 30.2 | 29.9 | 35.6 | 41.7 | 33.8 | 43.2 | 40.2 | 39.7 | 57.2 | 45.9 | 54.3 | 52.5 |
| DIVERSIFY | 47.7 | 32.9 | 44.5 | 31.6 | 30.4 | 37.4 | 48.7 | 46.9 | 49.0 | 59.9 | 51.1 | 67.6 | 55.0 | 62.5 | 61.7 |

USC-HAD, and PAMAP datasets. In each dataset, we randomly select four pairs of persons where one is the training and the other is the test. For simplicity, we use $0, 1, \cdots$ to denote different domains. All experiments are done in three random trials, mitigating the influence of unfair splits.

Table 2 and Table 3 show the results on four settings for HAR[3]. For all settings for HAR, our method achieves the best average performances. For Cross-person, Cross-position, Cross-dataset, and One-Person-To-Another settings, our method significantly outperforms the second-best baseline by $2.4\%$, $1.4\%$, $9.9\%$, and $5.8\%$ respectively. All results demonstrate the superiority of DIVERSIFY in different generalization settings.

We observe more insightful conclusions. (1) *When the task is difficult:* In the Cross-Person setting, USC-HAD may be the most difficult task. Although it includes more samples, it contains 14 subjects with only two sensors on one position, which may bring more difficulty in this situation. The experimental results prove the above argument that all methods perform terribly on this benchmark while ours has the largest improvement. (2) *When datasets are significantly more diverse:* Compared to Cross-Person and Cross-Position settings, Cross-Dataset may be more difficult since all datasets are totally different that samples are influenced by subjects, devices, sensor positions, and some other factors. In this setting, our method is substantially better than others. (3) *Limited data:* Compared with Cross-Person setting, One-Person-To-Another is more difficult since it has fewer data samples. Therefore, in this situation, enhancing diversity can bring a remarkable improvement and our method can boost the performance.

## 3.4 ANALYSIS

**Ablation Study**   We present ablation study to answer the following three questions: (1) *Why obtaining pseudo domain labels with class-invariant features in step 3?* If we obtain pseudo domain labels with common features, domain labels may have correlations with class labels, which may introduce contradictions when learning domain-invariant representations and lead common performance. This is certified by the results in Figure 4(a). (2) *Why using fine-grained domain-class labels in step 2?* If we utilize pseudo domain labels to update the feature net, it may make the representations seriously biased towards domain-related features and thereby leads to terrible performance on classification, which is proved in Figure 4(b). If we only utilize class labels to update the feature net, it may make representations biased to class-related features, make DIVERSIFY unable to obtain true latent sub-domains, and thereby has unremarkable performance, which is proved in Figure 4(c).

---

[3]In One-Person-To-Another setting, we only report average accuracy of four tasks on each dataset. Since only one domain exists in training dataset for this setting, DANN and CORAL cannot be implemented here.

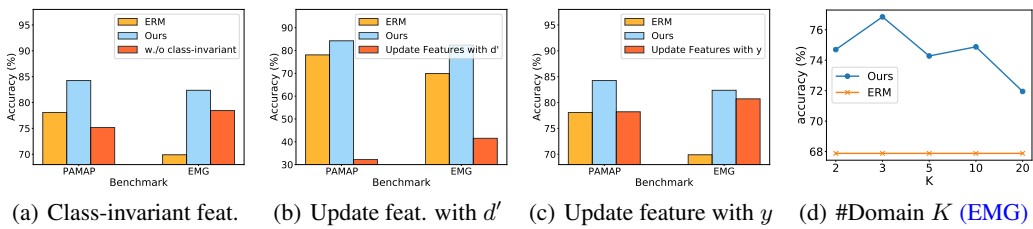

(a) Class-invariant feat.   (b) Update feat. with $d'$   (c) Update feature with $y$   (d) #Domain $K$ (EMG)

Figure 4: Ablation study.

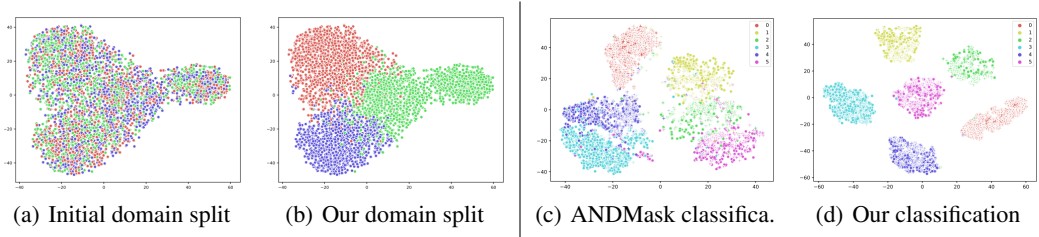

(a) Initial domain split   (b) Our domain split   (c) ANDMask classifica.   (d) Our classification

Figure 5: t-SNE visualizations for domain splits ((a) (b)) and classification ((c) (d)) on EMG data.

Therefore, we should utilize fine-grained domain-class labels to obtain representations with both domain and class information. (3) *The more latent sub-domains, the better?* More latent sub-domains may not bring better results as shown in Figure 4(d). A dataset may only contain a few sub-domains and introducing more may have a conflict with the intrinsic data property. Plus, more latent sub-domains also brings difficulty to obtain pseudo domain labels and learn domain-invariant features, which may have negative effects.

**Visualization Study**   We present some visualizations to show the rationales of our method and more results are in appendix. Data points with different initial domain labels are mixed together in Figure 5(a) while our DIVERSIFY can characterize different latent distributions to separate them well in Figure 5(b). It illustrates that our method can find better latent sub-domains. From Figure 5(d) and Figure 5(c), we can see that our method can learn better domain-invariant representations compared to the latest method ANDMask. Therefore, DIVERSIFY can find better latent sub-domains and then find domain-invariant representation from split domains, which can finally enhance generalization.

**Existence of latent distributions**   Latent sub-domains widely exist in time series, even in data collected from one person when performing the same activity. In Figure 6(a), for a subject in USC-HAD dataset, there is more than one latent distribution for his walking activities. Figure 6(b) demonstrates similar phenomena exist in EMG dataset where our algorithm found three latent distributions.

**Quantitative analysis for worst-case distributions**   We present quantitative analysis by computing the $\mathcal{H}$-divergence (Ben-David et al., 2010) to show the effectiveness of our 'worst-case distribution'. As shown in Figure 6(c) and Figure 6(d), compared with initial domain splits, latent sub-domains generated by our method have larger $\mathcal{H}$-divergence among each other. According to Prop. 2.1, larger $\mathcal{H}$-divergence among domains can bring better generalization.

## 4   RELATED WORK

**Time series classification** is a challenging problem for years. Existing researches mainly focus on the modeling of temporal relations using either RNN-based methods or the recently-proposed Transformer architecture. To our best knowledge, there is only one recent work (Du et al., 2021) that studied time series from the distribution level. However, their method is not end-to-end trainable since they developed a two-stage method that is non-differential.

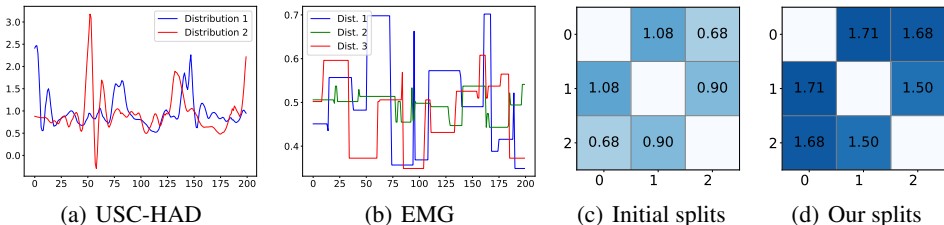

Figure 6: (a) (b) Latent distributions obtained by our approach on two datasets. (c) (d) $\mathcal{H}$-divergence among domains with initial splits and our splits on PAMAP.

**Domain / OOD generalization** (Wang et al., 2021a) mainly consists of data augmentation (Zhang et al., 2018; Yue et al., 2019), domain-invariant feature learning (Muandet et al., 2013; Ganin et al., 2016), and meta-learning (Li et al., 2018) techniques. The work of (Matsuura & Harada, 2020) also studied DG without knowing the domain labels by clustering with the style features for image data. However, that work is not applied to time series and it is not end-to-end trainable due to clustering. Disentanglement (Peng et al., 2019; Wang et al., 2021b) tries to disentangle the domain and label information, but they assume access to domain information. **Single domain generalization** is similar to our problem setting that also involves one training domain (Fan et al., 2021; Li et al., 2021; Qiao et al., 2020; Wang et al., 2021c). Most work adopted generative models and data augmentation strategies. However, they treated the single domain as *one* distribution and did not explore the latent sub-distributions. Specifically, Wang et al. (2021c) proposed a 'learning-to-diversify' algorithm that has a similar name to ours, but focused on mutual information maximization.

**Mixture models** (Rasmussen et al., 1999) are models representing the presence of subpopulations within an overall population, e.g., Gaussian mixture models. Our approach belongs to this category in general with focus on OOD. Moreover, we do not rely on generative models as most mixture models. **Subpopulation shift** is a new setting introduced in (Koh et al., 2021), which refers to the situation that the training and test domains overlap, but their relative proportions differ. Our problem does not belong to this setting since we assume that the training and test distributions do not overlap.

**Distributionally robust optimization** (Delage & Ye, 2010) shares a similar paradigm with our work, whose paradigm is also to seek a distribution that has the worst performance within a range of the raw distribution. However, we study the internal distribution shift instead of seeking a global distribution close to the original one. GroupDRO (Sagawa et al., 2020) studied the DRO at a group level, which is significantly different from our idea.

**Pseudo-labeling** is a common strategy in semi-supervised and unsupervised learning. SHOT (Liang et al., 2020) combined the pseudo-labeling strategy and IM (Information Maximization) to learn target adaptation features for source free domain adaptation. However, we mainly focus on domain generalization and thereby have different goals and updating styles. **DANN** (Ganin et al., 2016) has become the fundamental technology in domain adaptation and domain generalization, using adversarial training to obtain domain-invariant representations. In our method, we utilize it to obtain class-invariant and domain-invariant representations based on all features.

## 5 Conclusions and Future Work

We proposed DIVERSIFY to learn generalized representations for time series classification. DIVERSIFY employs an adversarial game that maximizes the worst-case distribution scenario while minimizing their distribution divergence. We demonstrated its effectiveness in different types of applications via qualitative and quantitative analysis. We are surprised that not only a mixed dataset, but one dataset from a single person can also contain several latent distributions. Characterizing such latent distributions will greatly improve the generalization performance on unseen datasets.

In the future, we plan to extend our algorithm by seeking new architectures and developing techniques to estimate the value of domain numbers $K$. Moreover, we plan to apply our algorithm to other time series-related applications such as forecasting and video classification.

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

# A    METHOD DETAILS

## A.1    DOMAIN-INVARIANT REPRESENTATION LEARNING

Domain-invariant representation learning utilizes adversarial training which contains a feature network, a domain discriminator, and a classification network. The domain discriminator tries its best to discriminate domain labels of data while the feature network tries its best to generate features to confuse the domain discriminator, which thereby obtains domain-invariant representation. Therefore, it is an adversarial process, and in our setting, it can be expressed as following,

$$
\begin{aligned}
\min_{h_b^{(4)}, h_c^{(4)}} \quad & \mathbb{E}_{(\mathbf{x},y)\sim\mathbb{P}^{tr}} \ell(h_c^{(4)}(h_b^{(4)}(h_f^{(4)}(\mathbf{x}))), y) - \ell(h_{adv}^{(4)}(h_b^{(4)}(h_f^{(4)}(\mathbf{x}))), d'), \\
\min_{h_{adv}^{(4)}} \quad & \mathbb{E}_{(\mathbf{x},y)\sim\mathbb{P}^{tr}} \ell(h_{adv}^{(4)}(h_b^{(4)}(h_f^{(4)}(\mathbf{x}))), d').
\end{aligned}
\tag{9}
$$

To optimize Eq. 9, we need an iterative process to optimize $h_b^{(4)}, h_c^{(4)}$ and $h_{adv}^{(4)}$ iteratively, which is cumbersome. It is better to optimize $h_b^{(4)}, h_c^{(4)}$ and $h_{adv}^{(4)}$ at the same time. It is obvious that the key is to solve the problems caused by the negative sign in Eq. 9. Therefore, a special gradient reversal layer (GRL), a popular implementation of the adversarial training in training several domains as suggested by (Ganin et al, 2016), came. GRL acts as an identity transformation during the forward propagation while it takes the gradient from the subsequent level and changes its sign before passing it to the preceding layer during the backpropagation. During the forward propagation, the GRL can be ignored. During the backpropagation, the GRL makes the sign of gradient on $h_b^{(4)}$ reverse, which solves the problems caused by the negative sign in Eq. 9.

# B    DATASET

## B.1    DATASETS INFORMATION

Table 4 shows the statistical information on each dataset.

Table 4: Information on HAR datasets.

| Dataset | Subjects | Sensors | Classes | Samples |
|---------|----------|---------|---------|---------|
| EMG     | 36       | 1       | 7       | 33,903,472 |
| SPCMD   | 2618     | -       | 35      | 105,829 |
| DSADS   | 8        | 3       | 19      | 1,140,000 |
| USC-HAD | 14       | 2       | 12      | 5,441,000 |
| UCI-HAR | 30       | 2       | 6       | 1,310,000 |
| PAMAP   | 9        | 3       | 18      | 3,850,505 |

## B.2    DATA PREPROCESSING

We will introduce how we preprocess data and the final dimension of data for experiments here. We mainly utilize the sliding window technique, a common technique in time-series classification, to split data. As its name suggests, this technique involves taking a subset of data from a given array or sequence. Two main parameters of the sliding window technique are the window size, describing a subset length, and the step size, describing moving forward distance each time.

For EMG, we set the window size 200 and the step size 100, which means there exist $50\%$ overlaps between two adjacent samples. We normalize each sample with $\tilde{\mathbf{x}} = \frac{\mathbf{x}-\min \mathbf{X}}{\max \mathbf{X}-\min \mathbf{X}}$. $\mathbf{X}$ contains all $\mathbf{x}$. The final dimension is $8 \times 1 \times 200$.

For Speech Commands, we follow (Kidger et al., 2020).

Now we give details on all datasets in Cross-person setting. For DSADS, we directly utilize data split by the providers. The final dimension shape is $45 \times 1 \times 125$. $45 = 5 \times 3 \times 3$ where 5 means five positions, the first 3 means three sensors, and the second 3 means each sensor has three axes. For USC-HAD, the window size is 200 and the step size is 100. The final dimension shape is

$6 \times 1 \times 200$. For PAMAP, the window size is 200 and the step size is 100. The final dimension shape is $27 \times 1 \times 200$. For UCI-HAR, we directly utilize data split by the providers. The final dimension shape is $6 \times 1 \times 128$.

For Cross-position, we directly utilize samples obtained from DSADS in Cross-person setting. Since each position corresponds to one domain, a sample is split into five samples in the first dimension. And the final dimension shape is $9 \times 1 \times 125$.

For Cross-dataset, we directly utilize samples obtained in Cross-person setting. To make all datasets share the same label space and input space, we select six common classes, including WALKING, WALKING UPSTAIRS, WALKING DOWNSTAIRS, SITTING, STANDING, LAYING. In addition, we down-sample data and select two sensors from each dataset that belong to the same position. The final dimension shape is $6 \times 1 \times 50$.

For One-Person-To-Another, we randomly select four pairs of persons from DSADS, USC-HAD, and PAMAP respectively. Four tasks are $1 \rightarrow 0, 3 \rightarrow 2, 5 \rightarrow 4$, and $7 \rightarrow 6$. Each number corresponds to one subject. And the final dimension shape is $45 \times 1 \times 125, 6 \times 1 \times 200$, and $27 \times 1 \times 200$ for DSADS, USC-HAD, and PAMAP respectively.

As we can see, samples in EMG and HAR are all have more than one channels (the first dimension shape), which means they are all multivariate.

### B.3 INITIAL DOMAIN SPLITS

We introduce how we split data here.

Since Speech Commands is a regular task, we just randomly split the entire dataset to a training dataset, a validation dataset, and a testing dataset.

We mainly focus on EMG and HAR, and we construct domains for OOD tasks. We denote subjects of a dataset with $0 - s_n$, where $s_n$ is the number of subjects in the dataset. For example, there are 36 subjects in EMG and we utilize $0, 1, 2, \cdots, 35$ to denote data of them respectively.

Table 5 shows the initial domain splits of EMG and all datasets for HAR in Cross-person setting. We just want make each domain has the similar number of samples in one dataset. As noted in the main paper, we also utilize 0, 1, 2, and 3 to represent different domains but they have different meanings with subjects. When conducting experiments, we take one domain as the testing data and the others as the training data. Our method is not influenced by the splits of the training data since we do not need the domain labels.

Table 5: Initial domain splits.

| Dataset | 0 | 1 | 2 | 3 |
|---|---|---|---|---|
| EMG | 0-8 | 9-17 | 18-26 | 27-35 |
| DSADS | 0,1 | 2,3 | 4,5 | 6,7 |
| USC-HAD | 0,1,2,11 | 3,5,6,9 | 7,8,10,13 | 4,12 |
| PAMAP | 2,3,8 | 1,5 | 0,7 | 4,6 |

## C NETWORK ARCHITECTURE AND HYPERPARAMETERS

For the architecture, the model contains two blocks, and each has one convolution layer, one pooling layer, and one batch normalization layer. A single-fully-connected layer is used as the bottleneck layer while another fully-connected layer serves as the classifier. All methods are implemented with PyTorch (Paszke et al., 2019). The maximum training epoch is set to 150. The Adam optimizer with weight decay $5 \times 10^{-4}$ is used. The learning rate for GILE is $10^{-4}$. The learning rate for the rest methods is $10^{-2}$ or $10^{-3}$. (For new added Speech Commands with MatchBoxNet3-1-64, we also try the learning rate, $10^{-4}$.) We tune hyperparameters for each method.

For the pooling layer, we utilize MaxPool2d in PyTorch. The kernel size is $(1, 2)$ ad the stride is 2. For the convolution layer, we utilize Conv2d in PyTorch. Different tasks have different kernel sizes and Table 6 shows the kernel sizes.

Table 6: The kernel size of each benchmark.

| EMG | (1,9) | | DSADS | (1,9) |
|---|---|---|---|---|
| SPEECHCOMMANDS | (1,9) | Cross-Person | USC-HAD | (1,6) |
| Cross-position | (1,9) | | PAMAP2 | (1,9) |
| Cross-dataset | (1,6) | One-Person-To-Another | The same as Cross-Person | |

# D    EVALUATION METRICS

We utilize average accuracy on the testing dataset as our evaluation metrics for all benchmarks. Average accuracy is the most common metric for DG and it can be computed as the following,

$$Acc = \frac{\sum_{(\mathbf{x},y)\in\mathcal{D}^{te}} I_y(y*)}{\#|\mathcal{Y}^{te}|},$$

$$y* = \arg\max h(\mathbf{x}). \tag{10}$$

$I_y(y*)$ is an indicator function. If $y = y*$, it equals 1, otherwise it equals 0. $\#|\cdot|$ represent the number of the set. $h$ is the model to learn. Please note that $\mathcal{X}^{te}$ has a different distribution from $\mathcal{X}^{tr}$ for EMG and HAR. And $\mathbf{x}$ has been preprocessed and each sample is a segment.

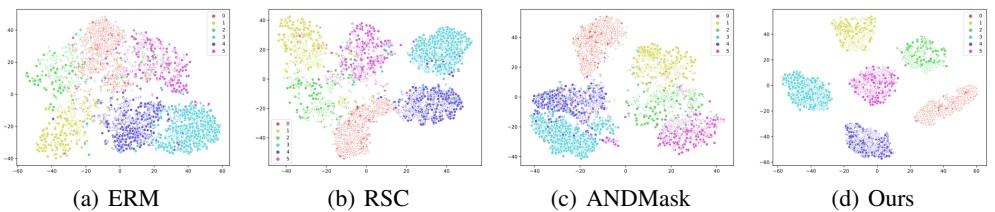

(a) ERM  (b) RSC  (c) ANDMask  (d) Ours

Figure 7: Visualization of the t-SNE embeddings for classification on EMG. Different colors correspond to different classes while different shapes correspond to different domains.

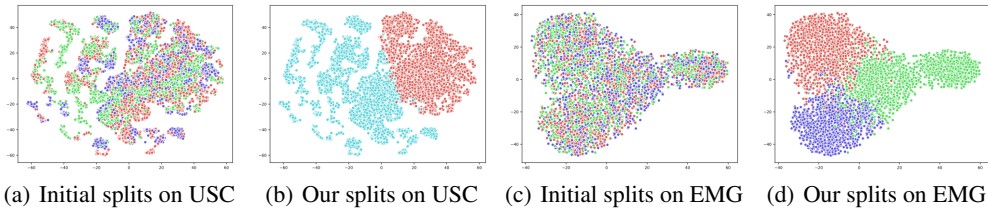

(a) Initial splits on USC  (b) Our splits on USC  (c) Initial splits on EMG  (d) Our splits on EMG

Figure 8: Visualization of the t-SNE embeddings for domain splits where different colors represent different domains.

# E    MORE EXPERIMENTAL RESULTS

## E.1    VISUALIZATION STUDY

We show more visualization study in this part. As shown in Figure 7(a) and Figure 7(b), both ERM and RSC also cannot obtain fine domain-invariant representations and our method still achieves the best domain-invariant representations. As shown in Figure 8, compare with initial domain splits, latent sub-domains generated by our method are better separated.

## E.2    PARAMETER SENSITIVITY

There are mainly four hyperparameters in our method: $K$ which is the number of latent sub-domains, $\lambda_1$ for the adversarial part in step 3, $\lambda_2$ for the adversarial part in step 4, and local epochs and total

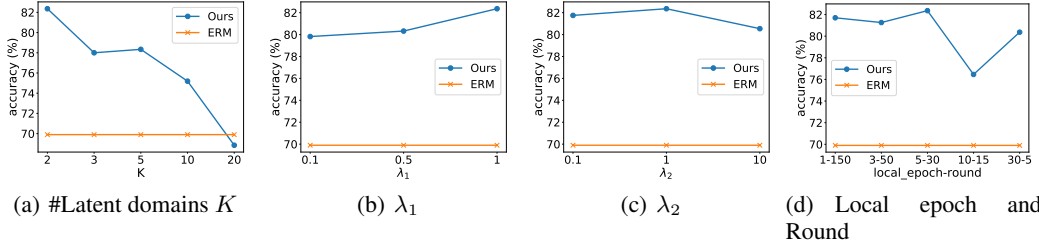

Figure 9: Parameter sensitivity analysis (EMG).

rounds. For fairness, the product of local epochs and total rounds is the same value. We evaluate the parameter sensitivity of our method in Figure 9 where we change one parameter and fix the other to record the results. From these results, we can see that our method achieves better performance in a wide range, demonstrating that our method is robust.

## E.3 $\mathcal{H}$-DIVERGENCE

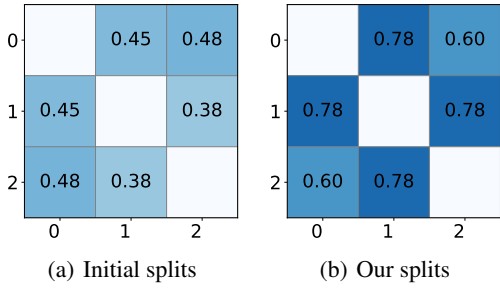

Figure 10: $\mathcal{H}$-divergence among domains with initial splits and our splits on EMG.

Figure 10 shows $\mathcal{H}$-divergence among domains with initial splits and our splits on EMG, which demonstrates our splits have larger $\mathcal{H}$-divergence and thereby can bring better generalization.

## E.4 THE INFLUENCE OF ARCHITECTURES

Table 7: Results on EMG dataset with different model sizes.

| Model Size | Small | | | | | Medium | | | | | Large | | | | |
| Target | 0 | 1 | 2 | 3 | AVG | 0 | 1 | 2 | 3 | AVG | 0 | 1 | 2 | 3 | AVG |
|---|---|---|---|---|---|---|---|---|---|---|---|---|---|---|---|
| ERM | 56.6 | 65.7 | 65.3 | 61.8 | 62.3 | 62.6 | 69.9 | 67.9 | 69.3 | 67.4 | 61.2 | 78.8 | 68.8 | 64.6 | 68.4 |
| DANN | 65.3 | 69.3 | 63.6 | 62.9 | 65.3 | 62.9 | 70.0 | 66.5 | 68.2 | 66.9 | 63.0 | 72.7 | 69.4 | 68.5 | 68.4 |
| CORAL | 66.9 | 74.9 | 70.8 | 73.2 | 71.4 | 66.4 | 74.6 | 71.4 | 74.2 | 71.7 | 67.7 | 77.0 | 72.7 | 71.8 | 72.3 |
| Mixup | 56.8 | 61.0 | 68.1 | 67.2 | 63.2 | 60.7 | 69.9 | 70.5 | 68.2 | 67.3 | 66.3 | 81.1 | 71.2 | 69.6 | 72.0 |
| GroupDRO | 64.9 | 75.0 | 71.6 | 69.1 | 70.1 | 67.6 | 77.5 | 73.7 | 72.5 | 72.8 | 66.3 | 79.3 | 74.9 | 71.3 | 73.0 |
| RSC | 62.7 | 73.2 | 67.6 | 64.0 | 66.9 | 70.1 | 74.6 | 72.4 | 71.9 | 72.2 | 65.1 | 76.8 | 72.2 | 67.7 | 70.4 |
| ANDMask | 62.4 | 66.0 | 66.3 | 65.6 | 65.1 | 66.6 | 69.1 | 71.4 | 68.9 | 69.0 | 65.7 | 78.1 | 72.1 | 71.9 | 71.9 |
| DIVERSIFY | 69.8 | 77.3 | 74.4 | 74.4 | 74.0 | 71.7 | 82.4 | 76.9 | 77.3 | 77.1 | 72.0 | 86.6 | 78.5 | 78.9 | 79.0 |

To ensure that our method can work with different sizes of models, we add some more experiments with more complex or simpler architectures. As shown in Table 7, where small, medium, large indicates the different model sizes (our paper uses the medium), we see a clear picture that model sizes influence the results and our method also achieves the best performance. Small corresponds to the model with one convolutional layer, Medium corresponds to the model with two convolutional layers, and Large corresponds to the model with four convolutional layers. For most methods, more complex models bring better results. For all architectures, our method achieves the best performance.

### E.5 RESULTS ON WESAD

We show results on a larger dataset, WESAD, here.

Wesad (Schmidt et al., 2018) is a publicly available dataset for wearable stress and affect detection, which contains physiological and motion data of 15 subjects. Wesad contains 63,000,000 instances. And we utilize techniques mentioned above to preprocess it. We utilize sensor modalities of chest-worn device including electrocardiogram, electrodermal activity, electromyogram, respiration, body temperature, and three axis acceleration. We split 15 subjects into four domains, [(0-3), (4-7), (8-11), (12-14)]. The results are shown in Table 8, and we can see that our method achieves the best performance compared to other state-of-the-art methods with an improvement over **8**%.

Table 8: Results on WESAD dataset.

| Target | 0 | 1 | 2 | 3 | AVG |
|---|---|---|---|---|---|
| ERM | 36.4 | 38.9 | 42.7 | 67.2 | 46.3 |
| GroupDRO | 44.3 | 58.3 | 43.2 | 64.1 | 52.5 |
| ANDMask | 42.5 | 45.3 | 51.2 | 62.8 | 50.5 |
| DIVERSIFY | **46.5** | **67.4** | **57.5** | **73.2** | **61.1** |

