# OpenReview forum: "DIVERSIFY to Generalize: Learning Generalized Representations for Time Series Classification"
_ICLR.cc/2022/Conference — ICLR 2022 Submitted_

### Official Review · Reviewer_8pYF · 2021-11-02

**Correctness:** 2
**Technical Novelty And Significance:** 3
**Empirical Novelty And Significance:** 2
**Recommendation:** 6
**Confidence:** 4

**Main Review:**

This is an interesting paper and potentially the start to something really interesting. There are a number of questions/concerns I have as described below. Sorry if this comes off as overly negative. I enjoyed reading the paper and thought this is interesting work.

Contributions/Arguments/Writing:
* There is a lot of discussion on stationarity early in the paper, with claims seeming to indicate that all time-series signals are non-stationary and other modalities, like images, are stationary. Strictly speaking, non-stationarity is a problem with practically all real-world classification problems, including images and other areas that "distributional" and "domain" work has been done. Perhaps the argument here should be around user-specific effects (e.g., the way one person wears the Myo EMG band may make gestures look different than another person? Or the same person wearing the device on two different days may require different calibration?).
* One thing I didn't understand was why "viewing time-series classification from a distributional perspective," at least using the methods in this paper, is important. Perhaps the authors can address this? Based on my cursory understanding of related work like GroupDRO, their goal is to minimize worst-case errors on specific distributions of data. You could imagine uses in ML fairness where you want the worst case performance for minority classes to be sufficiently good. However, the distributions used in this paper are latent and seemingly don't have semantic meaning.
* Given the distributional slant, are the metrics described the this paper the right ones for the task? (caveat: as noted below, I don't believe the metrics are named). Results are provided holistically for all distributions whereas perhaps we should be more curious to see results on specific distributions. Based on the theory it seems like worst case performance should be improved.
* The claims around what is or isn't a domain were not consistent between modalities. In the image modality (Fig 1a) a sketch is one domain whereas a cartoon is another. However, in Fig 1b it seems that two subsequences within one series are considered separate domains?

Models / algorithms / experiments
* The models used (as described in the appendix) are very small, especially for the Speech Commands dataset. It would be interesting to see state of the art (or at least recent) architectures for each problem (e.g., MatchBox or similar on Google Speech Commands) and to apply the paper's adversarial algorithm to those models. Based on a quick search it seems that this [1] is state of the art on this dataset achieves 97.41% compared to around 95% in this paper.
* Metrics: It would be useful to describe the metrics being used for each experiment. I can try to infer (I assume it's mostly per-clip accuracy?) but I don't know for sure. More broadly it's unclear how metrics are compute computed, whether or not they're the "right" metrics for each dataset, or if they're the right metrics for evaluating distributional approaches like the one described in the paper.
* It's hard to understand how substantial performance improvements are, especially on the EMG and Human Activity datasets. I could only find one other recent paper using the EMG dataset and performance was in the 90%s [2]. Assumably these are not apples-to-apples comparisons with the results in this paper... but having some external validation would be appreciated (especially on Speech Commands).
* Re: Ablation study. I don't know what dataset these experiments are being performed on. Is there a reason the performance decreases with more domains? I was surprised by the visualizations in Fig 8 and Fig 9. These seem to be contrary to what is described in the rest of the paper: it seems that the domains actually don't do a good job of capturing latent structure? Perhaps I'm reading these incorrectly.
* The EMG and HAR datasets are very small. I suppose this is philosophical, but would it be more impactful to develop algorithms that better fit this data or to collect 10x more data from each set of sensors? It's hard to know how well performance generalizes on datasets like this.

[1] Mordido et al. "Compressing 1D Time-Channel Separable Convolutions using Sparse Random Ternary Matrices." May 2021.
[2] Rubio et al. "Identification of hand movements from electromyographic signals using Machine Learning" 2020.



**Summary Of The Paper:**

This paper focuses on time-series classification problems, such as gesture or speech command recognition, where the data for any given sequence contains information from one user, but for which a priori you don't know anything about that user. Their goal is to be able to train on data with one set of users ("domains") and evaluate on a different set of users ("domains"). The users (at train and test) are unknown and are modeled by discrete latent variables.

Contributions include an algorithm ("DIVERSIFY") for learning distributions of domains/users. Their approach ultimately results in a model for time-series classification and for latent segmentation of data into discrete domains (aka user profiles). The model is framed using a min-max adversarial formulation.

The paper documents results on three datasets (EMG signals, Speech Commands, and Human Activities) show improvements over other distributional algorithms.

**Summary Of The Review:**

I could be swayed to change my ratings fro "correctness" and "empirical novelty" but in general there are too many unanswered questions for me to vote to accept this paper.

---

> ### Author Response · Authors · 2021-11-15
> **Response for reviewer 8pYF (part 1/3)**
>
> Thank you for your professional and constructive feedback. Now we answer your concerns below and we also revised our paper accordingly (see the updated version with blue texts).
>
> 1.**Non-stationary.**
>
> First of all, we are *not* saying that image domain has no non-stationary problem since the non-stationary property is a common problem widely existed in many applications such as images and time series.
>
> Second, our focus is on time series classification. Although there is much work on non-stationary image classification from the context of domain generalization / OOD, we argue that there is little work focusing on time series data.
> The biggest difference between time series data and image data is that traditional work on DG / OOD for image data require the domain label for each sample *known*, while it is often impossible to acquire such domain labels for a time series data that dynamically changes. (See Figure 1)
>
> Third, why is it so hard to acquire domain labels for time series? We agree with you that user-specific or environment-specific characteristics may be a reason for such difficulty: different users have different habits to wear a device or perform an activity; they can wear it on different positions; they can perform activities in different environments; etc.
> We see that such non-stationary property naturally exists in time series data. Thus, it becomes a challenging problem to design a generalized time series model.
>
>
> 2.**Why declaring modeling time series from a distributional perspective?**
>
> We use the expression "distributional perspective" to help better explain the rationale of our approach.
> In our problem, we see each segment of data is points sampled from a mixture distribution where each component of distribution corresponds to a domain.
> We try to maximize mixture distribution gaps in an implicit way (Step 3 with pseudo-labeling and an adversarial loss in our method. The worst case is our goal which may not be achieved but we can see that our splits indeed enlarge distribution gaps from Figure 6(c)-(d)).
> Our goal is to find these components and obtain common generalized features.
> It is obvious that the problem becomes a traditional DG problem after we exploit the mixture distribution and find sub-domains.
> Therefore, we design DIVERSIFY.
>
> Comparison with GroupDRO: GrouoDRO is to minimize worst-case errors on specific distributions of data and GroupDRO designs an uncertainty set of distributions to find the worst one.
>
> Moreover, in THEORETICAL INSIGHTS (Section 2.4 in the main paper), we explain why DIVERSIFY works from a distributional perspective.
> Once obtaining the worst-case distribution scenario, we can maximize the segment-wise distribution gap to preserve diversities and thereby our method can perform well on a wider range of distributions, which means we can have a good performance even on out-of-distribution data.
>
> Finally, we agree with you that seeking the semantic meaning of different distribution scenarios is important to help us understand the benefits and disadvantages of our approach.
> Currently, our focus is to learn *generalized* representations for better OOD performance, thus our approach may not be specifically designed to solve a particular type of semantic ML fairness.
> The worst-case distribution scenario can be seen as an *implicit* semantic version that we do not know about but can contribute to better generalization ability in the end.
> We do believe it is quite interesting to find out the semantic meaning behind our method given enough application-specific contexts and labels in the future.
>
> 3.**Are the metrics described in this paper the right ones for the task?**
>
> First, we mainly use classification accuracy as the evaluation metric, which follows existing endeavor on domain generalization / OOD prediction (Wang et al., 2021a) and DomainBed (Gulrajani and Lopez-Paz, 2021).
> We add more descriptions about it in our future version (appendix D).
> For time series, it is per-clip (segment) accuracy.
>
> Second, we explain more about metrics.
> We treat each segment as a sample and each segment has a corresponding label.
> And then we compute average accuracy on samples (segments) in testing.
> Average accuracy on a testing dataset that has a different distribution from the training dataset is the most common metric for DG.
> For time-series classification, most work often treats a part of data in a window (segment) as a sample and computes average accuracy on samples in testing.
>
> Third, we do not follow GroupDRO to test the worst-case accuracy since this is not our focus. The goal of DIVERSIFY is to learn generalized representations for time series for better OOD performance. Thus, we directly measure its accuracy on OOD domain, which is adopted by most DG/OOD works. But for the worst-case scenario, our results in Figure 6 (c) (d) show that our model can identify the worst-case scenario (with larger $\mathcal{H}$-divergence) for better generalization performance.

---

> ### Author Response · Authors · 2021-11-15
> **Response for reviewer 8pYF (part 2/3)**
>
> 4.**What is a domain?**
>
> Generally speaking, a domain is a collection of data samples that belong to the same probability distribution.
> The biggest difference between an image and a time series dataset is that an image can belong to only one domain in most circumstances.
> However, for a time-series sample, different segments may belong to different domains.
> For example, data collected via EMG may have different styles due to environmental or body changes even when doing the same activity in continuous time.
> It is the reason we want to perform DIVERSIFY for time-series classification.
>
> 5.**Results on Speech Command using the recommended backbone network.**
>
> For the results on speech command dataset, we use different model architectures from existing works. In fact, there is no standard models for speech command dataset. Therefore, to answer your question on the results, we replace the base feature net using two blocks of CNN with MatchBoxNet3-1-64 as you suggested.
> The results are shown in the table below.
>
> | Backbone  | 2-CNN | MatchBoxNet3-1-64 |
> |-----------|-------|-------------------|
> | ERM       | 93.64 | 96.73             |
> | GroupDRO  | 93.92 | 96.73             |
> | ANDMask   | 93.55 | 96.83             |
> | [1]       | -     | 97.21             |
> | Diversify | **94.97** | **97.38**             |
>
> Since data preprocessing, learning rate, learning schedule, and some other factors may be different from the implementations in [1], ERM only achieves 96.73%, which is slightly worse than the results in [1].
> However, our method still achieves the best performance, and it is also better than the results in [1], which demonstrates the superiority on regular time classification tasks of our method. (We believe our method can obtain better results if we finetune the basic hypermeters carefully.)
> Since the baseline is already very strong, it is difficult to make a remarkable improvement.
>
> As for the results of our paper using our own models, we implement all methods with the same architecture for fairness.
> Relative improvements can already demonstrate the superiority of our method since our methods perform better than GroupDRO, ANDMask, and some other state-of-the-art methods.
>
> To sum up, both two different models have shown that our method is agnostic to models and can achieve the best performance.
> More results about the effects of model architectures can be found in **Experiments with different model sizes.** of response to Review 7NdQ.
>
>
> 7.**How are substantial performance improvements?**
>
> Note that the experimental setting in [2] of your review comments are significantly different from ours which makes it not feasible to compare their absolute performance:
> - In [2], training data and testing data share the same distribution, and it can obtain better accuracy easily.
> - However, in our setting, training data and testing data have *different* distributions, which is more challenging.
>
> To understand how well our method is, we can see relative improvements.
> For example, compared to ERM on EMG, GroupDRO improves 5.4% while our method improves 9.7%. Since they are operating on the same architecture and have gone through extensive hyperparameter tuning, this shows that our method can achieve significantly better performance.
>
> 8.**Details on ablation study.**
>
> We will add details about what dataset these experiments are being performed on in our future version.
> Figure 4(d) is based on EMG.
>
> > Performance increase or decrease with domain $K$: why?
>
> If $K$ is too small, domain splits will be coarse and there will be too few sub-domains to obtain differences of different style data.
> If $K$ is too large, domain splits will be meticulous and there will be few samples in one domain, which makes it hard to exploit generalized representations across sub-domains.
> It is just like clustering methods that more clusters are not always better.
>
> We revised the reference of Figure 7 and Figure 8 in our updated version.
> We remove the figures with a few points and show the complete versions directly.
> Figure 8 shows visualizations of domain splits on USC-HAD and EMG where different colors mean different splits.
> In Figure 7, different colors mean different classes and different shapes mean different domains.
> Figure 7(a)-(c) are based on initial splits.

---

> ### Author Response · Authors · 2021-11-15
> **Response for reviewer 8pYF (part 3/3)**
>
> 9.**Performance on a larger dataset.**
>
> The datasets for HAR in our paper are popular and relatively large datasets (i.e., PAMAP2 and USC-HAD) with 5M samples.
> We evaluate our method in a larger dataset, WESAD, that contains 63,000,000 samples, which is 10x larger than USC-HAD.
> The results are shown in the table below.
> Our method still achieves the best performance and has an over **8%** improvement compared to GroupDRO. Details on this dataset are presented in the last part of the appendix.
>
> | Target   | 0    | 1    | 2    | 3    | AVG  |
> |----------|------|------|------|------|------|
> | ERM      | 36.4 | 38.9 | 42.7 | 67.2 | 46.3 |
> | GroupDRO | 44.3 | 58.3 | 43.2 | 64.1 | 52.5 |
> | ANDMask  | 42.5 | 45.3 | 51.2 | 62.8 | 50.5 |
> | Diversify  | **46.5** | **67.4** | **57.5** | **73.2** | **61.1** |
>
> If you think these responses address your concerns, please consider increasing your score. Thank you!

---

> ### Comment · Reviewer_8pYF · 2021-11-21
> **Response to authors**
>
> Thanks to the authors for spending significant effort on the rebuttal. This was clearly a lot of work.
>
> After reading through all of the responses I still have mixed feelings. On one hand this approach does seem to improve performance. On the other hand I’m not convinced by the “distributional” and adversarial focus of the paper. I think there is a much simpler slant focused around doing simple clustering on the data which doesn’t requires any of the proofs or understanding of ideas that they build on (e.g., pseudo labeling). The distributional focus made the paper much more difficult to understand. It is used as a key motivator but there isn’t sufficient verification that the distributions mean anything substantive. So again, it “works,” but the paper doesn't do justice to "why" is works.
>
> My key gripes at this point are with how the paper is presented and some of the claims.
> * I appreciate that the authors added a “contributions” section at the beginning, however, unless I’m misunderstanding, I actually think the first claim (“novel problem”) is wrong. This isn’t the first time people have thought about distributions on time-series data or “generalized representations.” The authors cite recent working doing the same (e.g., Du 2021) and there are likely tens if not hundreds of other papers doing the same across areas of time-series machine learning (computer vision, ubiquitous computing, bio sensing, etc). If the intent here is that the authors use latent variables for the first time to identify this distribution, then that is also wrong. I don’t want to cite myself but I published on something like this on time-series data in the early/mid 2010s… and there are surely many more earlier references.
> * In the rebuttal section #4 (“What is a domain”) I disagree with the statement: “The biggest difference between an image and a time series dataset is that an image can belong to only one domain in most circumstances” Images commonly contain many distributions. Imagine an image with 5 people in it, each performing different actions. Each person may be sampled from a different distribution, their actions may be of different distributions, etc. If you are doing semantic segmentation then you have to deal with non-stationarity from both a signal processing and semantic level. On a related note, in general the paper is written “computer-vision-first” when it’s really a “time-series-first” paper. This might make sense for CVPR but less so for ICLR?
> * Re: rebuttal section #9 (larger dataset): When it comes to time-series data, the raw number of samples doesn’t really mean much. 5M samples sampled at 1000 hz could mean 1.3 hours of data. 5M samples at 100 hz means 13 hours of data. In practice, WESAD is sampled at 700 hz and others at 100 hz. Thinking of this like “how many gestures/commands” and “how many user sessions” are in the dataset is much more meaningful. I appreciated the results on WESAD, but is has roughly the same number of users compared to the other non-speech datasets (PAMAP2 has 9 users; USC-HAD has 14; WESAD has 15).
>
> I’ll up my vote to borderline-accept but would like for some of these claims to be revised for final submission.

---

> > ### Author Response · Authors · 2021-11-22
> > **Thank you for your response**
> >
> > We are glad that you acknowledge our response. While we are extremely thankful that you increased the score, we would like to make some further clarifications to some of your remaining concerns below.
> >
> > 1. More explanations about the **distributional view.**
> >
> > Currently, our focus is to learn generalized representations for better out-of-distribution performance. Thus, we mainly want to discover the latent *distributions* that could be utilized for better generalization (Ben-David et al., 2007).
> > For this purpose, it is important to learn domain-invariant representations (survey by Wang et al., 2021), which is often achieved by adversarial training like DANN (Ganin et al., 2016).
> >
> > - Clustering? On the other hand, the clustering technique may seem to be a feasible solution, but it focuses on *instance-wise* distances rather than *distribution-wise* divergence. Thus, clustering may work, but not sufficient to learn distribution-invariant representations.
> > - Why it works? We made two kinds of explanations in the paper: (1) the theoretical motivation in section 2.4 well aligns with our method and motivation; (2) apart from experimental results, we also provided sufficient visualizations (Fig. 5, 6, 7, 8, and 10) to show that our algorithm is able to learn the latent distribution information contained in the time series data.
> >
> > 2. **Clarifications on problem setting.**
> >
> > Compared with AdaRNN (Du et al., 2021), our work is significantly different from three aspects:
> > - First, AdaRNN focuses on time series forecasting problems while our is on classification.
> > - Second, and the most significant difference, AdaRNN utilizes a two-stage method that is non-differential compared with our end-to-end solution.
> >
> > Then, for the claim of 'fist time', we revised it to avoid such an improper claim. Our main contribution to problem setting is to identify the generalized time series classification for OOD settings.
> >
> > 3. More explanations on **what is a domain**?
> >
> > For image data in the domain generalization setting, we mean that image *classification* (i.e., determine which category an image belongs to) often involves one domain, not a general image in other CV tasks such as segmentation. This claim is from the survey work of (Wang et al., 2021) and DomainBed (Gulrajani and Lopez-Paz, 2021) that most of the work on DG/OOD is for general image classification task which treat one image instance as one domain. Sorry we did not make it clear in the paper.
> > Domain splits for image classification are based on image styles.
> > An image with 5 people in it, each performing different actions, may have multiple labels, which may be different from ours.
> >
> >
> > Thank you for your professional and constructive feedbacks again. We will further revise our paper carefully in the final version.

---

> > > ### Comment · Reviewer_8pYF · 2021-11-22
> > > **Response to authors**
> > >
> > > The following are a couple random thoughts from your response. No need to respond. I'm just thinking out loud. :)
> > >
> > > Perhaps I'm wrong, but don't standard classification models typically learn domain-invariant representations assuming they're trained on large enough and diverse enough data (e.g., with thousands of people in the Speech Commands dataset)? This is one reason why I'm not convince that the distribution aspect is what's improving things here. I don't think that the visualizations fully validate that. I don't know if there is even a way to fully validate given that domains in this case are ambiguous and the same person can have data belonging to multiple domains. Given that everything is latent we may never know whether or not domains are "properly" assigned during training (nor is there even a necessarily a "proper" assignment). Again, maybe I'm thinking of this all wrong...
> > >
> > > In my opinion the fact that image classification is sometimes perceived as being single-domain is a moot point here. Even if that task is single-domain, the problems discussed in this paper are unrelated.
> > >
> > > Anyway... nice paper and thanks for the back and forth.

---

### Official Review · Reviewer_7NdQ · 2021-11-02

**Correctness:** 3
**Technical Novelty And Significance:** 2
**Empirical Novelty And Significance:** 3
**Recommendation:** 6
**Confidence:** 3

**Main Review:**

### Strength
1. This paper extended the domain generalization on time series classification problem by modeling the distribution of sub-domains within each domain and demonstrate its benefit in learning better representation.
2. The proposed approach shows superior performance on three different benchmarks compared to state-of-the-art domain generalization methods. Multiple generalization settings are provided in human activity recognition task and the improvement is consistent across different settings.


### Weakness
1. The overall technical novelty is marginal. First, the pseudo-labeling strategy is adopted directly from (Liang et al. 2020). Secondly, the adversarial training strategy is adopted from (Ganin et al., 2016). Finally, the determination of number of sub-domains is based on simple heuristics i.e. cross-validation on different K values, which is not really a new concept compared to decoding time series using a state-space model such as hidden Markov model. The authors need to emphasize their contributions on existing development of domain generalization.
2. The theoretical insight is based on existing conclusion from (Sicilia et al., 2021). It does not add to the novel contribution of the paper other than providing an interpretation of loss function.
3. The discussion on classification model is not sufficient. Is the same model architecture as described in Appendix B used for all datasets and experiments? What is the impact of using more complex or simpler architectures? The ablation study can be strengthened by comparing different choices of model artitechtures.


### Additional Comments
1. It would be good to elaborate more implementation details such as how data is pre-processed, how convolution is performed for multi-variate time series, what is the kernel size, and what is the criteria for convergence in repeating step 2-4. The discussion on model architecture in Appendix B should be placed in main paper instead.
2. Figure 6(d) shows even when K=2, the accuracy is much higher than ERM. I'm wondering what is the performance when K=1, which can be considered as degenerated baseline of the proposed approach. This can provide additional evidence on how effective is characterizing sub-domains and promote diversity.

**Summary Of The Paper:**

This paper proposed a time series classification method with loss function that simultaneously promotes diverse distribution among different sub-domains within each domain and domain invariant feature representation within the same class. Here the domain is considered in a rather granular manner such as different person may be considered as different domains in gesture recognition application. An iterative process is proposed to learn a classification model. Both theoratical analysis and empirical evaluation on three different applications are provided. The proposed method shows better accuracy especially on generalizing across different domains than other competing methods.

**Summary Of The Review:**

Overall, the proposed approach showed quite promising results on several different time series classification applications. The problem formulation and techniques are mostly adopted from existing work, which reduce the technical novelty. The authors need better justification on the performance improvement achieved by proposed approach, and highlight their contributions in domain generalization methodology. Please refer to the main review for more suggestions on improvement.

---

> ### Author Response · Authors · 2021-11-15
> **Response to reviewer 7NdQ (part 1/2)**
>
> Thank you for your professional and constructive feedback. Now we answer your concerns below and we also revised our paper accordingly (see the updated version with blue texts).
>
> 1.**Technical novelty: what's the difference between ours and existing work?**
>
> First of all, we want to emphasis that our work is to learn the generalized representation for time series data, which is a new and challenging problem (as recognized by you and other two reviewers). Thus, we turn to using some existing common algorithms to see if they can solve the problem.
> In fact, these two techniques (pseudo-labeling and adversarial training) are all common basic methods that have been utilized in much existing work.
> We will show some slight differences of these techniques in our methods and rewrite the related work section to emphasis our differences.
>
> (i) The pseudo-labeling strategy.
>
> It is a common and simple pseudo-labeling strategy inspired by clustering.
> SHOT (Liang et al. 2020) combines the pseudo-labeling strategy and IM (Information Maximization) to learn target adaptation features without changing the source classifier for source-free domain adaptation problems.
> However, in our method, we combine the pseudo-labeling strategy and adversarial loss to learn class-invariant features without changing the basic feature network to obtain sub-domains.
> The goals, updating styles, and design styles are all different.
>
> (ii) The adversarial training.
>
> Adversarial training, or GAN (Goodfellow et al., 2014), is one of the most popular basic models in both machine learning and domain generalization that has become the *fundamental* technology behind many existing methods. For instance, DANN (Ganin et al, 2016) is one of the earliest methods to apply adversarial training to domain adaptation.
> We utilize it to obtain class-invariant and domain-invariant features without changing the basic feature network.
>
> (iii) The determination of number of sub-domains $K$.
>
> We agree with you that $K$ is an important variable and currently, we treat it as a hyperparameter in the method.
> If $K$ is too small, domain splits will be coarse and there will be too few sub-domains to obtain differences of different distributions.
> If $K$ is too large, domain splits will be meticulous and there will be few samples in one domain, which makes it hard to exploit generalized (common) representations across sub-domains.
> Therefore, we set $K$ as a hyperparameter and utilize validation data to determine it.
> We also believe that $K$ can be somewhat pre-determined in the future work.
>
> **Our main contributions are shown in the following.**
>
> - Novel problem: We find the problem that different parts of one time-series sample may belong to different distributions and try to learn generalized representations to construct models that can perform well on out-of-distribution data.
> - Effective end-to-end solution: For the problem of *time-series classification* tasks *without domain labels*, we *formulate it* as a domain generation problem, *view it* from the distribution perspective, *propose DIVERSIFY* to solve it.
> - Good results: Our method achieves remarkable results on many benchmarks. To the best of our knowledge, no other existing work pays attention to learning generalized representations for time-series classification tasks without knowing domain labels.
>
> 2.**Theoretical insight.**
>
> We would like to emphasize that theoretical innovation is *not* the focus of the paper, but act as the fundamental pillar and proof of our method.
> We mainly want to propose a method to learn generalized representation for time series classification without domain labels.
> Our main contributions are exploiting sub-domains, learning generalized representation, and finally obtaining a model that can perform well on out-of-distribution data.
> Theoretical insight offers the rationality of our method from another perspective.

---

> ### Author Response · Authors · 2021-11-15
> **Response to reviewer 7NdQ (part 2/2)**
>
> 3.**Experiments with different model sizes.**
>
> First of all, existing experiments are fair since we use the same model architectures for each dataset and method to ensure fairness. For different datasets, the kernels' sizes may be different.
> To ensure that our method can work with different sizes of models, we add some more experiments with more complex or simpler architectures.
> As shown in the table below for EMG data, where *small, medium, large* indicates the different model sizes (our paper uses the medium), we see a clear picture that model sizes influence the results, and our method also achieves the best performance.
> Small corresponds to the model with one convolutional layer, Medium corresponds to the model with two convolutional layers, and Large corresponds to the model with four convolutional layers.
> For most methods, more complex models bring better results.
> For all architectures, our method achieves the best performance. (Corresponding to E.4 in appendix)
>
> | Model Size | Small |      |      |      |      | Medium |      |      |      |      | Large |      |      |      |      |
> |------------|:-----:|:----:|:----:|:----:|:----:|:------:|:----:|:----:|:----:|:----:|:-----:|:----:|:----:|:----:|:----:|
> | Target     | 0     | 1    | 2    | 3    | AVG  | 0      | 1    | 2    | 3    | AVG  | 0     | 1    | 2    | 3    | AVG  |
> | ERM        | 56.6  | 65.7 | 65.3 | 61.8 | 62.3 | 62.6   | 69.9 | 67.9 | 69.3 | 67.4 | 61.2  | 78.8 | 68.8 | 64.6 | 68.4 |
> | DANN       | 65.3  | 69.3 | 63.6 | 62.9 | 65.3 | 62.9   | 70.0 | 66.5 | 68.2 | 66.9 | 63.0  | 72.7 | 69.4 | 68.5 | 68.4 |
> | CORAL      | 66.9  | 74.9 | 70.8 | 73.2 | 71.4 | 66.4   | 74.6 | 71.4 | 74.2 | 71.7 | 67.7  | 77.0 | 72.7 | 71.8 | 72.3 |
> | Mixup      | 56.8  | 61.0 | 68.1 | 67.2 | 63.2 | 60.7   | 69.9 | 70.5 | 68.2 | 67.3 | 66.3  | 81.1 | 71.2 | 69.6 | 72.0 |
> | GroupDRO   | 64.9  | 75.0 | 71.6 | 69.1 | 70.1 | 67.6   | 77.5 | 73.7 | 72.5 | 72.8 | 66.3  | 79.3 | 74.9 | 71.3 | 73.0 |
> | RSC        | 62.7  | 73.2 | 67.6 | 64.0 | 66.9 | 70.1   | 74.6 | 72.4 | 71.9 | 72.2 | 65.1  | 76.8 | 72.2 | 67.7 | 70.4 |
> | ANDMask    | 62.4  | 66.0 | 66.3 | 65.6 | 65.1 | 66.6   | 69.1 | 71.4 | 68.9 | 69.0 | 65.7  | 78.1 | 72.1 | 71.9 | 71.9 |
> | DIVERSIFY  | **69.8**  | **77.3** | **74.4** | **74.4** | **74.0** | **71.7**   | **82.4** | **76.9** | **77.3** | **77.1** | **72.0**  | **86.6** | **78.5** | **78.9** | **79.0** |
>
>
> 4.**More implementation details.**
>
> > Pre-processing steps
>
> We preprocess data with sliding window and normalization by following existing work on the dataset introduction paper, which are common techniques to proprecess time series data.
>
> > Model architecture and validation metrics.
>
> For multi-variate time series data, we just treat each variate as one channel.
> The kernel sizes are different for different datasets, and we will add it in Appendix.
>
> > Convergence and model selection criteria
>
> Our model can converge in several rounds in real experiments. This is done by following DomainBed and utilizing validation to obtain the final model.
>
> > More details on model architecture
>
> We put it in both the main paper and appendix C to introduce more details on model architecture.
>
> 5.**$K=1$.**
>
> In fact, when $K=1$, our method equals to ERM (since there will be no latent sub-domains) and results showed our superiority.
>
> If the response addresses your concerns, please consider increasing the score. Thank you!

---

> > ### Comment · Reviewer_7NdQ · 2021-11-20
> > **Post-rebuttal review**
> >
> > The authors responded to all my questions. On the contributions of the paper, I agree with the authors that the main novelty is problem formulation and empirically showing diversify the representation learning improves recognition performance. On the other hand, the rebuttal also confirmed my belief that the technical novelty is limited due to the adoption of existing techniques in a straightforward way. Furthermore, the discussion on adversarial learning for domain generalization needs strengthening. For example, [1] utilize adversarial learning to improve generalization across subjects in action recognition application. More related work should be surveyed.
> > Overall, I feel this is still borderline work. Adoption and adjustment of existing methods may not be a fatal flaw as the empirical results showed significant improvement compared to other state-of-the-art. I appreciate the authors' effort in providing additional results so that I know the improvement is consistent across different model complexities. I'm willing to change my rating to weakly accept considering the novel problem formulation and strong empirical results. But I will not be upset if we choose to reject the paper.
> >
> > [1] R. Zhao, K. Wang, H. Su, Q. Ji, Bayesian graph convolution lstm for skeleton based action recognition, ICCV 2019

---

> > > ### Author Response · Authors · 2021-11-21
> > > **Further Response**
> > >
> > > We are glad that you acknowledge our response. Now, we elaborate on our technical novelty further.
> > >
> > > 1.**Differences between ours and [1]**
> > >
> > > (1) [1] is for **action recognition** based on the skeleton data which are often videos.
> > > In our experiments, we focus on **sensor-based activity recognition**.
> > > (2) [1] prefers a feature representation to be invariant of subject so that the model will not overfit to the nuisance caused by subject-dependent variation and it utilizes adversarial learning to achieve this goal.
> > > And in the paper, adversarial learning is mainly to regularize the model parameters.
> > > Our method is for **common time-series classification** where we even **do not know the domains (subjects)**.
> > >
> > > 2.**The technical novelty.**
> > >
> > > The technical novelty of our method is how to **adapt** and **combine** existing base models (methods) to this novel problem.
> > > We do not focus on proposing a new base model to solve the problem.
> > > In the following, we list some of our adaptations.
> > >
> > > (1) We propose **pseudo domain-class labels** to fully utilize the knowledge contained in the domains and classes and learn fine-grained information.
> > > (2) We adopt DANN, a most famous method for DA (Domain adaptation) and DG (Domain generalization), to obtain **class-invariant** features and combine self-supervised learning to obtain sub-domain labels.
> > > The traditional DANN and its applications are to obtain **domain-invariant** features.
> > > (3) Based on **fine-grained** features, we apply DANN to obtain domain-invariant features.
> > > (4) We **combine** all to solve the problem, learning the generalized representation for time series data, and introduce the **theoretical insight** behind it.
> > >
> > > 3.**Adversarial learning and generalization.**
> > >
> > > Adversarial learning is a common base method for domain generalization.
> > > Lots of methods utilize is for their problems.
> > > For example, [2] utilized adversarial learning and style clustering for domain generalization in image classification.
> > > It is better to pay attention to when and how to adapt it to the corresponding problems.
> > > And in our method, we adapt adversarial learning for this novel problem mentioned above.
> > >
> > > **As you said, "I'm willing to change my rating to weakly accept considering the novel problem formulation and strong empirical results."
> > > Please note that 6 corresponds to weak accept.**
> > > If the response addresses your concerns, please consider increasing the score.
> > > Thank you!
> > >
> > >
> > > [1] R. Zhao, K. Wang, H. Su, Q. Ji, Bayesian graph convolution lstm for skeleton based action recognition, ICCV 2019
> > >
> > > [2] Matsuura T , Harada T . Domain Generalization Using a Mixture of Multiple Latent Domains, AAAI 2020

---

> > > ### Author Response · Authors · 2021-11-22
> > > **Further response to review 7NdQ: Did our further response resolve your remaining concerns?**
> > >
> > > Dear reviewer, we thank you again for acknowledging our response. Did we resolve your remaining concerns? If not, do you have other concerns? We would like to resolve them; If so, would you mind increasing the score to 6, which is weak accept as stated in your last response?
> > >
> > > Many thanks for your efforts in helping to make this work better:)

---

### Official Review · Reviewer_kiW8 · 2021-11-05

**Correctness:** 4
**Technical Novelty And Significance:** 3
**Empirical Novelty And Significance:** Not applicable
**Recommendation:** 8
**Confidence:** 4

**Main Review:**

Strengths:
- Good representation.

- The motivation and model design makes sense to me.

- Extensive experiments. This work used 6 datasets from 3 different scenarios, worked on 4 different challenging tasks. I also appreciate the visualizations.


Weaknesses:

No obvious weakness to me. Here are some suggestions for further revision:

- This is an important point. Section 3.3, regarding cross-person generalization setting: I read the descriptions in the first set and noticed the footnote 2, but how to split 14  (USC-HAD) and 9 (PAMAP2) subjects into 4 groups? So I guess you are not splitting the subjects but splitting the segments, right? In detail, you shuffled all segments from all subjects, and split them into 4 groups (each group contains segments from all subjects).

I also understand that the term of 'cross-person' is used in some papers (especially in EEG related publications) referring to shuffling all segments from all subjects. However, in their case, 'cross-person' is in contrast to 'person dependent' (use the segments from the same subject for both training and test) and 'person independent' (use several subjects for training and others for test).

In your case, it's not good to call it 'cross-person generalization'. Because, in the context of 'cross-position' and 'cross-dataset' where the training set and test set are from different positions/datasets, so readers suppose 'cross-person' means the training set and test set are from different subjects. Thus, my suggestion is to change the setting from splitting segments to splitting subjects. For example, in PAMAP2, you can use 7 subjects for training and 2 residuals for test (it's kind of like setting 4 but not the same). At least, please provide the results of splitting subjects over one dataset.

- BTW, I didn't find the results for setting one-person-to-another (not in Table 2 nor Table 3), am I missed something?

- Please enlarge panel c of Fig 1, the point shapes are hardly readable.

- This paper takes univariate time series as an example, please discuss how to deal with multivariate time series? Specifically, are the signals from different sensors (in the same segment) belong to the same domain label? If not, how to solve the problem?

- Please add more descriptions and details of domain-invariant representation learning (bottom in Page 4 or add to appendix). It's not clear to me that why the GRL learned representation is invariant to domains?

- Please make Eq 1 an in-line equation. It's not important for the whole paper.

- Please clarify, are the h_f, h_b, h_c are all functions? Implemented by neural networks? Please let me know if I missed some related information in your experimental setup.

**Summary Of The Paper:**

This work proposes a framework to learn a domain-invariant representation of time series. Using the idea of domain generalization, this paper removes the influence of varying domains/distributions across patients/datasets through pseudo domain classification. Although this kind of model has been rather studied in images, but not a lot in time series.

**Summary Of The Review:**

This paper focuses on one meaningful topic, targets the challenge, and proposed an interesting solution to solve it. The presentation is easy to follow, the model is good (maybe the pseudo self-classification is not very novel, but still ok to me), and the experiments are sound to me. So I suggest accepting it.

---

> ### Author Response · Authors · 2021-11-09
> **Thank you for your feedback**
>
> Thank you for your professional and constructive feedbacks. Now we answer your concerns below with revised paper version.
>
>
> 1.**Data split and cross-person setting.**
>
> There may be some misunderstanding of the cross-person setting:
> Since we focus on learning generalized representations, different domains contain data of different persons *without* overlapping.
> For example, for USC in Cross-person setting, there are 14 persons in total.
> We denote 14 persons with 0-13 and split 14 persons into 4 groups: (0,1,2,11), (3,5,6,9), (7,8,10,13), and (4, 12).
> Why this split? Because we try to ensure each domain has similar number of data.
> When testing, we treat one of domains as the target and the others as training data.
> Therefore, we see that there is no overlap between the training and testing domains.
>
> 2.**Results for one-person-to-another experiments.**
>
> It is in Figure 4 of the main paper (lower right part of page 7). Now we cancel the figure and move it to the main table 3 in the revised version.
>
> 3.**Point size.**
>
> We will enlarge the points' size of figure 1 in our future version.
>
> 4.**Multivariate time series.**
>
> We are sorry that we didn't make it clear: in fact, we are using multivariate time series for the EMG and HAR experiments.
> Each sensor often contains data multiple axes and one signal contains multiple sensors.
> The signals from different sensors (in the same segment) belong to the same domain label.
> We will make modifications in the paper accordingly.
>
> 5.**Explanations of GRL layer on learning domain-invariant representations.**
>
>
> Domain-invariant representation learning utilizes adversarial training which contains a feature network, a domain discriminator, and a classification network.
> The domain discriminator tries its best to discriminate domain labels of data while the feature network tries its best to generate features to confuse the domain discriminator, which thereby obtains domain-invariant representation.
> The adversarial process can be implemented with a a special gradient reversal layer (GRL).
> GRL acts as an identity transformation during the forward propagation while it takes the gradient from the subsequent level and changes its sign before passing it to the preceding layer during the backpropagation.
> This is a popular implementation of the adversarial training in training several domains as suggested by (Ganin et al, 2016).
> We will add more descriptions and details of domain-invariant representation in our future version.
>
>
> 6.**Equation.**
>
> We will make Eq. 1 an in-line equation.
>
> 7.**Yes, $h_f$, $h_b$, $h_c$ are all functions implemented by neural networks.**

---

> > ### Comment · Reviewer_kiW8 · 2021-11-19
> > **Keep suggested score as 8: accept**
> >
> > I thank the authors for preparing the responses. My concerns are largely addressed and I will keep my score unchanged as 8.

---

### Author Response · Authors · 2021-11-09
**General response**

----Update * 2: We have updated the PDF again with slight modifications per reviewer 8pYf's further comments.

----Update: We have updated the PDF accordingly. We add some clarification with **blue** texts to help better express our idea.

We would like to sincerely thank all reviewers for your constructive feedbacks of our paper. We are glad that all of you agreed with the contributions, problem importance, and results of this work.

For your concerns, as the rebuttal period begins, we will do our best to add further experiments and settings according to your feedback and make modifications to the paper in the next days.

---

### Public Comment · ~Jindong_Wang1 · 2023-03-24
**Update**

We would like to express our gratitude to anyone who is interested in this work.

Just an update: this work has been accepted at ICLR 2023: https://openreview.net/forum?id=gUZWOE42l6Q.

Code: https://github.com/microsoft/robustlearn.

---

### Decision · Program_Chairs · 2022-01-20

**Decision:**

Reject

**Comment:**

This paper has been reviewed by three reviewers, two scoring it borderline leaning towards an accept, and one scoring it as accept. The key criticism from reviewers is the lack of novelty (*the technical novelty is limited due to the adoption of existing methods without substantial changes (pseudolabeling for latent distribution, adversarial learning for domain generalization)*) and limited technical analysis (*Theoretical analysis is weak due to the use of existing conclusion from Sicilia et al., 2021*). On the other hand, authors argue that the problem they address is new (*learn the generalized representation for time series data, which is a new and challenging problem*).

As it stands, AC sees this paper as borderline leaning towards reject for the above reasons even though the evaluations are interesting.

---

> ### Public Comment · ~Jindong_Wang1 · 2022-01-29
> **Response to the decision**
>
> **We are extremely confused by the final decision.**
>
> ICLR 22 has an acceptance rate of 30% and ours is rated as the top 13% (866, avg 6.67) of all submissions, **which is clearly not a "borderline" paper stated by ACs.**
>
> Strange things happen: We see that there are papers accepted with scores 683 this year. And also there are papers rejected with scores 10-8-6-5. I myself am also a reviewer for ICLR and I see that my reviewed paper got accepted with scores all 6.
>
> Maybe a rebuttal to the decision that is useless:
>
> We have clearly addressed the reviewer's concerns in the rebuttal. But every paper has its drawbacks, that's why we are not 10-10-10.
>
> Speaking of novelty, as we stated in response to reviewer 7NdQ: we are an application paper, and the idea of adversarial training and pseudo labeling are quite general. Most importantly, we are not a simple combination of them, but adaptations. This is generally favored by other two reviewers with a confidence of 4, while the other one is with confidence 3.
>
> With all due respect, this decision is not responsible to the reviewers and authors and we cannot accept it (but we have to). **If the reviews and scores are not important, then why not let ACs and PCs do the review directly?**
>
> Finally, we are also disappointed that PCs even do not bother to reply to our emails.